# Exploring Worldwide Wardrobes to Support Reuse in Consumers' Clothing Systems

**Dieuwertje de Wagenaar [1],\*, Joris Galama [2] and Siet J. Sijtsema [2]**

[1] Wageningen Corporate Value Creation, Circular Fashion Lab, 6700 HB Wageningen, The Netherlands
[2] Wageningen Economic Research, 6708 PB Wageningen, The Netherlands; joris.galama@outlook.com (J.G.); siet.sijtsema@wur.nl (S.J.S.)
\* Correspondence: dieuwertje.dewagenaar@wur.nl

**Abstract:** Extending the use of garments is often seen as an important strategy to decrease the impact of the fashion industry. However, currently there are a lack of data on and understanding of consumers' wardrobes. This study explores consumers' wardrobes internationally, and we aim to explore the total amount, unused and second-hand garments in order to develop interventions to support reuse. Through an online course, data were gathered in a survey about the content of participants' wardrobes, counting the amounts of garments in predefined categories, and the amount of unused and second-hand garments thereof. Differences were found between clothing categories, age groups and gender for unused and second-hand garments. Between nationalities only differences were found for second-hand garments. These insights are supportive to targeted interventions for gender and age groups related to specific categories of (unused and second-hand) garments, to elongate the practical service life of garments, support consumers' sustainable clothing decisions and in the end reduce consumption. Additionally, this exploration provides insights how to improve international monitoring and the value of digital wardrobe studies. Recommendations are provided, especially focused on interventions to support motivations, capabilities, and opportunities to improve reuse. Ultimately, through consumers' wardrobes this study supports the next steps towards a more circular clothing system.

**Keywords:** wardrobe study; reuse; interventions; behavior change model; consumer; garments; second-hand; circular fashion; monitoring

## 1. Introduction

The number of garments that are produced, as well as purchased, used, and disposed has grown in the last few decades [1]. Fast fashion is seen as the main reason for this and has a large impact on the environment [2]. Fast fashion is caused by consumers' willingness to follow the latest trends and purchase the newest garments, which has caused an almost doubled number of garments that have been produced in the last 15 years [3]. This means that garments are being worn for a shorter amount of time (practical service life) than they could have been worn (technical service life) [4]. Furthermore, less than 1% of the materials being used to produce garments are recycled to make new garments [3]. Therefore, the current clothing system can be regarded as a (mainly) linear system, which is wasteful and polluting [3]. Changing business as usual to a circular economy is getting more and more attention nowadays [5]. The Ellen MacArthur Foundation describes the circular economy based on three main principles: "(1) Design out waste and pollution, (2) keep products and materials in use, (3) regenerate natural systems" [3] (p. 49). This is very much aligned with the so-called R-ladder, naming 10 R-strategies for circularity in order of most circular to least circular: (1) refuse, (2) rethink, (3) reduce, (4) reuse/resell, (5) repair, (6) refurbish, (7) remanufacture, (8) repurpose, (9) recycle, (10) recover/remine [6]. In essence, these strategies say that there are multiple ways to reduce the amount of resources which are

used to produce new products. Different R-strategies require fewer resources compared to other strategies. For example, "refuse" or "rethink" products require fewer resources compared to "recycling" [6]. Originally these strategies were formulated from a business and technical perspective, but Campbell-Johnston et al. also introduce the consumer for some of the strategies [7]. To support the circular system, both suppliers and consumers should be taken into consideration [8]. In below table we present what kind of consumer behavior with regard to textiles could be linked to the different R strategies. There is a need for this, since these principles and strategies for circularity can only be successful if consumers accept and act according to Camacho-Otero et al. [9]. Increasing consumers' awareness of the clothing lifecycle and its impact is the best hope for sustainability in fashion industry, as stated by Harris et al. based on expert perspectives [10]. Therefore, in this study we focus on the consumer (behavior) perspective of circular fashion, in which a division is made between buying, using and disposal. The hierarchy of the R-ladder in relation to consumer behavior according to the levels of circularity in relation to textiles and clothing is specified for more circular behaviors in Table 1.

**Table 1.** Consumer behavior in relation to textiles and clothing within the R-ladder. The higher the R-strategy, the more circular and sustainable.

| Behavior | Consumer Behavior | R-Strategy [6,7] |
|---|---|---|
| Buy | Buying less | Refuse |
| | Buying second-hand<br>Buying high quality timeless items | Rethink |
| | Buying items with recycled materials<br>Swapping instead of buying | Reduce |
| Use | Reusing (second-hand) items<br>Better washing practices<br>Leasing/renting garments<br>Selling/donating garments for reuse | Reuse/resell |
| | Repair of garments<br>Careful use | Repair |
| | Changing garments to better suit style/size | Refurbish |
| | Upcycling garments to new garments | Remanufacture |
| Dispose | Upcycling garments to other purpose than clothing | Repurpose |
| | Disposal/donation for recycling | Recycle |
| | Disposal in household waste | Recover/re-mine |

With regard to the fashion industry different circular activities are already taking place. ThredUp [11] shows that there is an increase in consumers who are willing to buy, for example, second-hand clothes and environmentally friendly brands. "Consumers who prefer to buy from environmentally friendly brands", increased from 57% in 2013 to 72% in 2018 [11] (p. 11). First, these were mainly the early adaptors, and now it is visible that millennials and Gen Z are more and more adopting secondhand apparel compared to other age groups [11]. Furthermore, it is expected that the amount of second-hand clothes will make up one third of wardrobes in 2033 [11]. Nevertheless, in order to turn these relatively positive predictions into real behavioral change, more data and better understanding are needed about current behavior in relation to (un)used garments and second-hand items and interventions to support consumers.

What garments people have, how much, and how they are related to each other is often investigated by wardrobe studies [12]. The wardrobe is merely a place to store garments and other items; however, it is also a system with several "push" and "pull" factors which

determine the inflow and outflow of clothes in a wardrobe [12]. This view corresponds with a quote from Klepp and Bjerck: "these frames refer not only to the physical walls of the closet, but also to an entire structure of different storage spaces with corresponding criteria for where and what clothes should be kept and how clothes should be moved between them" [12] (p. 375). However, buying new items is not always combined with an outflow of garments [13,14]. Therefore, the in- and outflows have consequences for total wardrobe counts, and subsequently for growth or shrinkage thereof. Furthermore, different kinds of feedback can help to balance the in- and outflow of the wardrobe, for example by knowing what is already inside the closet or having experience with (un)used garments.

Harris et al. state that to create changes in consumers' purchase, use and disposal behavior, a focus on sustainability alone will not be sufficient for three reasons: (i) clothing sustainability is too complex; (ii) consumers are too diverse in their ethical concerns; (iii) clothing is not an altruistic purchase [10]. In their study, they applied a multistakeholder approach and defined interventions targeting suppliers, buyers, retailers and consumers, which encourage more sustainable clothing production, purchase, care and disposal behavior. This underlines the importance of the view of the wardrobe as a system. In this study, the wardrobe will be investigated as a static system to gain insight into the current state of wardrobes worldwide, and their (un)used and second-hand garments. Unfortunately, this study is not able to cover the entire inflow and outflow of the wardrobe, but specifically will focus on the use phase of consumers and related R-strategy, "Reuse". Based on this exploration, recommendations are given about possible interventions to move the practical service life closer to the technical service life of garments.

However, in order to reach behavior change by means of interventions to support circular fashion, a better and more holistic understanding of behavior change is needed. A useful model is the behavioral change wheel developed by Michie et al. [15]. They recognize that behavior is part of an interacting system involving capabilities (knowledge and skills), opportunities (social and physical context) and motivations (psychological reflective processes such as attitudes and beliefs), which need attention for successful interventions to change behavior. Besides the wheel, they also provide nine types of interventions: e.g., education, persuasion, incentivization, coercion, training, enablement, modelling, environmental restructuring and restrictions.

More specifically, Harris et al. proposed 15 interventions which are based on the perceived challenges and barriers of experts and of which eight are related to clothing purchasing, care and disposal [10]. Harris et al. and Markkula and Moisander note that informing and educating consumers of their actions will not be enough [10,16]. This is in line with Michie et al., who state that personal motivations (attitudes and beliefs), capabilities (knowledge and skills) and opportunities (social and physical environment) in everyday life practices of consumers should be aligned [15]. Table 2 shows interventions specified for improving reuse for consumers, such as "upcycling" or "leasing/hiring clothes", which in this study are considered as consumer behaviors. Therefore, specific interventions for elongating the life cycle of garments and extending their practical service life within the holistic system of the behavioral change wheel are still below par or simply lacking. Ultimately, this study aims to gain better understanding of consumers' wardrobes and their (unused)- and second-hand garments, to provide more insight in targeted reuse interventions and how these could be combined for potential impact.

**Table 2.** Sources of behavior in relation to consumer behavior interventions for textiles and fashion.

| Sources of Behavior [15] | Intervention Functions [15] | Interventions [10] |
|---|---|---|
| Motivation | Incentivization, Coercion, Modelling and Persuasion | **Communicate time, money and labor savings from reduced frequency and temperature of washing clothes.** Align buyers' and suppliers' remuneration with sustainability objectives. Accentuate benefits other than price to consumers to increase the value of their clothes. Normalize designs of sustainable clothing. Gain and maintain consumers' trust. |
| Capability | Education and Training | Make it easy for consumers to buy sustainable clothing. **Social marketing campaigns.** Involve designers in sustainability strategy. Improve transparency of supply chain. **Provide tools and assistance to help consumers understand their preferred style and cuts that suit their body shapes. Include textile skills in the school curriculum. Upcycling.** |
| Opportunity | Environmental restructuring, Enablement and Restrictions | **Retailers provide repair and recycle services. Leasing/hiring clothes. Legislate clothing recycling.** |

Note. Interventions related to R-strategy "reuse or resell" and consumer behavior are shown in bold.

## 2. Materials and Methods

In January 2020, Wageningen University and Research in collaboration with ArtEZ University of the Arts launched a Massive Open Online Course (MOOC) on Circular Fashion, Design, Science and Value in a Sustainable Clothing Industry [17]. As part of the online course, each participant was asked to fill out a small survey about the content of their wardrobe. Participation in the assignment was not based on reward. Filling out the assignment had no result on the final grade (pass or fail) on the online course. It was an ungraded assignment in the course design and participants gave informed consent to use the data in scientific research.

In total 606 (N = 606) have filled out the survey or at least a part of it. In order to only include respondents who filled out the survey seriously, we excluded respondents who only filled out up to 3 clothing categories, used incomprehensible ways to indicate the amount of clothes for each category (e.g., only mentioning "yes" or "no" or saying "20 + " items), and lastly owning less than 30 items. The latter is the amount of clothes that is often used as the minimum amount of clothes that someone needs, often defined as a "capsule wardrobe" or "minimalist wardrobe" [18,19]. The remaining sample consisted of 520 respondents (N = 520), of which 78.3% were female, 58.1% were aged between 18–30 years, 27.9% were between 31–50 years old and 5.6% were older than 51 years. Overall, the respondents were foremost from Europe (47.9%), followed by Asia (17.3%) and North- and South America (12.5 and 12.7%, respectively). An overview of the demographical information of the sample can be found in Table 3.

The methodological approach of conducting a "wardrobe study" is not new and can be performed in very different ways. Klepp and Bjerck explain a wardrobe study in the following way: "(...) we call it a wardrobe study, which allows for the analysis of the way in which clothes relate to each other on the whole or within parts of the wardrobe" [12] (p. 373). Wardrobe studies can have different goals and are used within different kinds of research fields (e.g., consumer studies, anthropology, sociology, marketing, etc.). For this study we only used self-reported quantitative data in order to get as many complete surveys as possible. According to the overview of methods that can be used for wardrobe studies, this method matches the "inventories/records wardrobe studies" the most, which is defined as: "Object-based research consists of the meticulous study and recording of objects' form, material, condition, and distribution" [12] (p. 379).

**Table 3.** Demographical information of respondents.

| Category | Number (%) |
|:---:|:---:|
| Gender | |
| Female | 407 (78.3) |
| Male | 85 (16.3) |
| Other/none | 28 (5.4) |
| Age | |
| 18–30 | 302 (58.1) |
| 31–50 | 145 (27.9) |
| 51+ | 29 (5.6) |
| Unknown | 44 (8.5) |
| Continent | |
| Africa | 11 (2.1) |
| Asia | 90 (17.3) |
| Europe | 249 (47.9) |
| North America | 65 (12.5) |
| Oceania | 12 (2.3) |
| South America | 66 (12.7) |

Respondents within the "Circular Fashion MOOC" were asked to fill out a short questionnaire about what is inside their wardrobe. After filling out information about nationality, age and gender, respondents were asked to fill out information about their wardrobe. In total we asked respondents to fill out 17 predefined clothing categories and one blank category so respondents could add one clothing category themselves. Furthermore, the respondents were asked to indicate how many of these items were unused items and/or second-hand. These questions are based on the wardrobe study by Maldini et al. [20]. The main difference in methodology is the way data were collected. Instead of visiting participants of the study and counting the items together, this study collected all data digitally by the participants themselves as an assignment within the online course.

Data was analyzed using IBM SPSS Statistics (version 25). After cleaning the data, as described in 2.1, descriptive information and differences between groups (age, gender and nationality) were calculated. This was completed for the 7 most popular clothing categories based on the top 5 (unused and second-hand) clothing categories of which the respondents indicated to have the most items of: short-sleeve T-shirts and tops, shoes and boots (pairs), shoes and boots (pairs), blouses and shirts, coats and jackets (including rain jackets and sport jackets), bags (only bags used as clothing accessories (excluding shopping bags) and dresses.

Differences between age categories and nationalities were calculated using the Hochberg GT2 post-hoc test, because of unequal sample sizes. Differences between males and females were calculated using an independent samples $t$-test [21]. Age was divided into the same three age categories as the study of Maldini et al. [20] (18–30, 31–50, 51+), and 8 nationalities were compared with each other (6 European countries with the highest count, and two non-European countries with the highest count (India and the United States). Lastly the total amount of clothes was measured using only the 17 predefined categories, in order to make the amounts comparable.

### 3. Results

*Descriptive Statistics*

In Table 4 the means are shown for each selected clothing category. Furthermore, the number of unused items and second-hand items are added, including the relative numbers. When adding up all the 17 predefined clothing categories, the respondents on average had an amount of 132.33 clothing items of which 32.91 (24.87%) were unused, and 32.83 (16.61%) were second-hand. The clothing categories of which the respondents overall had the most items were (1) short-sleeve T-shirts and tops, (2) shoes and boots (pairs) and

(3) sweaters and cardigans. When looking at the number of unused items, almost the same top three could be found: (1) short-sleeve T-shirts and tops, (2) shoes and boots (pairs), and (3) dresses. Lastly, for second-hand clothing items the respondents indicated to have the most items within the categories: (1) short-sleeve T-shirts and tops, (2) coats and jackets and (3) sweaters and cardigans. Nevertheless, the differences are rather small between the first four categories; only the category short-sleeve T-shirts and tops is overall the largest category.

When looking at the relative numbers, and the percentage of unused or second-hand clothes of the total amount of clothes within that category, it was found that scarves and shawls are the most likely to be unused (32.85%) and coats and jackets are the most likely to be second-hand (22.99%).

**Table 4.** Descriptive statistics of all clothing categories.

| | Minimum | Maximum | Mean | Std. Deviation | % of Total |
|---|---|---|---|---|---|
| Total number of clothes * | 30.00 | 713.00 | 132.33 | 84.390 | |
| Total number of unused clothes * | 0 | 406.00 | 32.91 | 46.867 | 24.87% |
| Total number of second-hand clothes * | 0 | 266.00 | 21.98 | 32.827 | 16.61% |
| Coats and jackets (including rain jackets and sport jackets) | 0 | 100 | 10.18 | 8.432 | |
| Of which unused | 0 | 40 | 2.44 | 3.929 | 23.97% |
| Of which second-hand | 0 | 28 | 2.34 | 3.612 | 22.99% |
| Shoes and boots (pairs) | 1 | 102 | 16.40 | 15.091 | |
| Of which unused | 0 | 70 | 4.29 | 7.586 | 26.16% |
| Of which second-hand | 0 | 60 | 1.72 | 4.537 | 10.49% |
| Bags (only bags used as clothing accessories; excluding shopping bags, for example) | 0 | 80 | 7.95 | 8.885 | |
| Of which unused | 0 | 50 | 2.41 | 5.134 | 30.31% |
| Of which second-hand | 0 | 40 | 1.38 | 3.204 | 17.36% |
| Scarves and shawls | 0 | 88 | 6.94 | 8.857 | |
| Of which unused | 0 | 51 | 2.28 | 5.061 | 32.85% |
| Of which second-hand | 0 | 30 | 1.19 | 3.059 | 17.15% |
| Hats | 0 | 40 | 2.92 | 3.473 | |
| Of which unused | 0 | 15 | 0.83 | 1.698 | 28.42% |
| Of which second-hand | 0 | 12 | 0.44 | 1.239 | 15.07% |
| Gloves (pairs) | 0 | 60 | 2.10 | 3.652 | |
| Of which unused | 0 | 60 | 0.59 | 2.882 | 28.10% |
| Of which second-hand | 0 | 25 | 0.29 | 1.427 | 13.81% |
| Suits | 0 | 91 | 1.81 | 4.831 | |
| Of which unused | 0 | 15 | 0.45 | 1.458 | 24.86% |
| Of which second-hand | 0 | 11 | 0.28 | .861 | 15.47% |
| Trousers | 0 | 50 | 7.54 | 6.634 | |
| Of which unused | 0 | 40 | 1.95 | 3.679 | 25.86% |
| Of which second-hand | 0 | 15 | 1.30 | 2.537 | 17.24% |
| Jeans | 0 | 60 | 6.25 | 5.235 | |
| Of which unused | 0 | 30 | 1.38 | 2.654 | 22.08% |
| Of which second-hand | 0 | 13 | 0.84 | 1.815 | 13.44% |
| Shorts (including sportswear) | 0 | 50 | 6.31 | 5.595 | |
| Of which unused | 0 | 18 | 1.39 | 2.722 | 22.03% |
| Of which second-hand | 0 | 15 | 0.78 | 1.732 | 12.36% |
| Sweaters and cardigans | 0 | 65 | 10.52 | 8.649 | |
| Of which unused | 0 | 31 | 2.29 | 4.092 | 21.77% |
| Of which second-hand | 0 | 30 | 2.15 | 3.945 | 20.44% |
| Short-sleeve T-shirts and tops | 0 | 150 | 18.70 | 14.970 | |
| Of which unused | 0 | 70 | 4.32 | 7.319 | 23.10% |
| Of which second-hand | 0 | 45 | 2.87 | 5.548 | 15.35% |
| Long-sleeve T-shirts and tops | 0 | 50 | 8.31 | 7.372 | |
| Of which unused | 0 | 40 | 1.64 | 3.477 | 19.74% |
| Of which second-hand | 0 | 40 | 1.30 | 3.152 | 15.64% |
| Blouses and Shirts | 0 | 70 | 10.30 | 10.372 | |
| Of which unused | 0 | 39 | 2.28 | 4.506 | 22.14% |
| Of which second-hand | 0 | 40 | 2.00 | 4.092 | 19.42% |

**Table 4.** *Cont.*

|  | Minimum | Maximum | Mean | Std. Deviation | % of Total |
|---|---|---|---|---|---|
| Dresses | 0 | 100 | 9.71 | 11.064 | |
| Of which unused | 0 | 80 | 2.71 | 6.130 | 27.91% |
| Of which second-hand | 0 | 39 | 1.79 | 3.908 | 18.43% |
| Jumpsuits | 0 | 23 | 1.56 | 2.557 | |
| Of which unused | 0 | 15 | 0.35 | 1.148 | 22.44% |
| Of which second-hand | 0 | 15 | 0.23 | 0.966 | 14.74% |
| Skirts | 0 | 40 | 4.83 | 5.338 | |
| Of which unused | 0 | 25 | 1.32 | 2.623 | 27.33% |
| Of which second-hand | 0 | 20 | 1.07 | 2.243 | 22.15% |

Note. * = sum of 17 predefined clothing categories.

As can be found in Table 5 and 6, differences were calculated between gender, age categories and countries. First, differences were found when comparing males and females. Significant differences were found between the (1) total of number of clothing items, (2) total amount of unused clothing items, (3) total number of second-hand clothing items, (4) (unused and second-hand) shoes and boots, (5) (unused and second-hand) sweaters and cardigans, (6) (unused and second-hand) blouses and shirts, (7) coats and jackets, (8) (unused and second-hand) bags and (9) (unused and second-hand) dresses. Overall, females had more items in all these categories.

When looking at age categories, some differences can be found between the age categories 18–30, 31–50 and 51+. On average respondents older than 31, have more clothes compared to respondents in the age category 18–30. Regarding unused clothing items, only differences can be found between the age categories 18–30 and 51+, of which the latter have the most unused clothing items. Furthermore, differences are found for the categories: shoes and boots, sweaters and cardigans, and dresses. For all these categories nearly the same results are found, namely that the older respondents have the most clothing items within each clothing category. For more detailed information see Table 5.

**Table 5.** Differences between gender and age categories.

| Clothing Category | Gender | | Age | | |
|---|---|---|---|---|---|
| | Female (N = 407) | Male (N = 85) | 18–30 (N = 302) | 31–50 (N = 145) | 51+ (N = 29) |
| Total number of clothes * | 142.30 (87.58) ** | 87.16 (44.02) ** | 124.00 (72.93) [a] | 147.57 (95.65) [b] | 175.21 (123.41) [b] |
| Total number of unused clothes * | 35.98 (50.09) ** | 18.98 (22.37) ** | 28.78 (37.76) [a] | 39.77 (58.82) [a,b] | 54.24 (73.98) [b] |
| Total number of second-hand clothes * | 24.60 (34.57) ** | 10.93 (21.33) ** | 22.62 (32.70) | 20.72 (30.54) | 26.86 (50.80) |
| Short-sleeve T-shirts and tops | 19.04 (15.47) | 16.96 (11.13) | 18.68 (15.69) | 18.41 (14.23) | 18.72 (12.65) |
| Of which unused | 4.40 (7.34) | 3.89 (5.58) | 3.99 (6.83) | 4.91 (8.80) | 5.41 (7.01) |
| Of which second-hand | 3.08 (5.64) | 2.08 (5.62) | 3.10 (5.82) | 2.26 (4.92) | 2.66 (5.27) |
| Shoes and boots (pairs) | 17.61 (15.20) ** | 11.47 (12.36) ** | 14.39 (11.72) [a] | 20.57 (17.73) [b] | 23.59 (23.94) [b] |
| Of which unused | 4.82 (8.28) ** | 2.34 (3.55) ** | 3.63 (6.65) [a] | 5.69 (9.45) [b] | 6.79 (9.55) [a,b] |
| Of which second-hand | 1.95 (4.89) ** | 0.88 (3.01) ** | 1.70 (4.86) | 1.86 (3.82) | 2.48 (6.44) |
| Sweaters and cardigans | 11.43 (9.07) ** | 6.36 (4.96) ** | 9.90 (7.95) [a] | 11.62 (8.98) [a,b] | 14.59 (13.54) [b] |
| Of which unused | 2.55 (4.45)** | 1.22 (1.95) ** | 2.08 (4.05) | 2.69 (4.49) | 3.07 (3.95) |
| Of which second-hand | 2.40 (4.08) ** | 0.78 (1.91) ** | 2.22 (3.43) | 1.99 (4.50) | 2.07 (4.80) |
| Blouses and Shirts | 10.91 (10.75)* | 7.89 (8.47)* | 9.79 (9.82) | 11.91 (12.26) | 11.1 (8.73) |
| Of which unused | 2.44 (4.59) ** | 1.35 (2.84) ** | 1.92 (3.85) | 2.95 (5.78) | 3.45 (4.90) |
| Of which second-hand | 2.22 (4.19)* | 1.21 (3.93)* | 2.31 (4.71) | 1.66 (2.91) | 1.66 (3.93) |
| Coats and jackets (including rain jackets and sport jackets) | 10.66 (8.84)* | 8.21 (6.38)* | 9.87 (8.92) | 10.98 (7.85) | 11.41 (8.21) |
| Of which unused | 2.58 (4.16) | 1.68 (2.62) | 2.30 (4.00) | 2.63 (4.22) | 2.76 (2.50) |
| Of which second-hand | 2.48 (3.54) | 1.92 (4.25) | 2.59 (3.96) | 1.97 (2.89) | 2.83 (4.77) |
| Bags (only bags used as clothing accessories; excluding shopping bags, for example) | 8.93 (9.23) ** | 3.12 (2.93) ** | 7.37 (8.03) | 9.21 (9.66) | 10.76 (11.96) |
| Of which unused | 2.79 (5.55) ** | 0.82 (1.79) ** | 2.31 (4.93) | 2.99 (6.16) | 3.14 (4.84) |
| Of which second-hand | 1.58 (3.46) ** | 0.48 (1.35) ** | 1.34 (3.09) | 1.36 (2.77) | 2.38 (6.14) |
| Dresses | 11.80 (11.39) ** | 0.71 (3.49) ** | 9.22 (9.55) | 11.49 (14.51) | 11.34 (11.11) |
| Of which unused | 3.25 (6.61) ** | 0.38 (3.25) ** | 2.18 (3.70) [a] | 3.72 (9.41) [b] | 4.48 (8.35) [a,b] |
| Of which second-hand | 2.23 (4.29) ** | 0.07 (0.55) ** | 1.78 (3.97) | 2.08 (4.37) | 1.83 (3.01) |

Note. * = $p < 0.05$; ** = $p < 0.01$, significant differences between age groups are indicated with [a], [b] meaning $p < 0.05$.

Lastly the total amount of (unused and second-hand) clothing items were compared between six European countries, India, and the United States. Overall, it was found that there were no differences between the selected countries when looking at the total amount of clothes and total amount of unused clothes. However, differences were found when looking at the total amount of second-hand clothes. French and Indian respondents on average had less second-hand clothes compared to Dutch, American and British respondents. German, Spanish and Italian respondents did not significantly differ from any other country. For more details see Table 6.

**Table 6.** Differences between nationalities.

| Category | France (N = 59) | The Netherlands (N = 45) | The United Kingdom (N = 25) | Spain (N = 20) | Germany (N = 18) | Italy (N = 16) | India (N = 52) | The United States (N = 31) |
|---|---|---|---|---|---|---|---|---|
| Total amount of clothes * | 99.68 (48.20) | 136.7 (98.47) | 130.4 (70.97) | 128 (60.10) | 143.94 (74.03) | 148.66 (64.51) | 120.98 (71.65) | 144.61 (79.51) |
| Total amount of unused clothes * | 23.83 (25.85) | 28.62 (23.81) | 29.72 (42.11) | 29.65 (47.42) | 34.61 (33.55) | 31.56 (29.80) | 24.38 (28.31) | 37.71 (40.64) |
| Total amount of second-hand clothes * | 10.02 (12.75) [a] | 37.31 (47.08) [b] | 34.56 (47.78) [b] | 14.65 (15.69) [a,b] | 28.33 (31.32) [a,b] | 15.69 (26.23) [a,b] | 10.13 (15.99) [a] | 36.32 (39.94) [b] |

Note. * = $p < 0.05$; Significant differences between nationalities are indicated with [a], [b] meaning $p < 0.05$.

## 4. Discussion

### 4.1. Exploring Wardrobes

The respondents owned, on average, 132.33 clothing items, of which 32.91 (25%) were unused, and 21.98 (17%) were second-hand. Interestingly, this is below the average found by Maldini and her colleagues in their study in 2017. On average the total amount of clothing items in their study was 173 clothing items, presenting quite a difference. Several reasons can be found for this. First, Maldini et al. have included data of two clothing categories, socks and underwear, based on a Euromonitor dataset [20]. In this study these categories were excluded. As this research is investigating possibilities and effectiveness of an online monitoring wardrobe study by building datasets effectively and internationally, using different data sources was not the aim for this study. Second, the respondents in this study participated by filling in their answers individually through a digital survey without supervision, whereas Maldini et al. visited the homes of their participants to count every clothing item in their home [20]. It could be argued that the latter could be more rigorous and less susceptible to errors or forgotten items. Third, as in this study the respondents fill in the survey without supervision, errors could be made in counting the total number of items of each category. Additionally, the respondents must make decisions on which clothing item fits in which predefined clothing category in the survey. This leaves room for interpretation and could result in varying interpretations between respondents and therefore incomparable results. Lastly, the study of Maldini et al. [20] was based on a non-representative sample of fifty Dutch participants, where this study is based on non-representative international respondents. The international differences could play a role in the total clothing items; think, for example, of the importance of differences between results of participants of only high-income countries or also low-income countries as included in this study. However, more importantly in this context, the original survey as designed by Maldini et al. did not allow for culturally defined clothing categories [20]. Therefore an 18th category was included in the survey to fit this study. This category was left blank so that respondents could add any clothing categories that were missing based on their individual wardrobe. Unfortunately, the results in this category were of such difference (ranging from pajamas to socks, lingerie, swimwear, etc.), that it was not considered in the data analysis. Any clothing items relevant, for example, to cultural expression that were included in the responses in this category were left out and might have resulted in a lower total average of clothing items.

Regarding the different clothing categories congruent to the wardrobe study of Maldini et al., respondents indicated to have the most items within the categories which can be defined as upper wear (e.g., T-shirts, sweaters, and blouses and shirts) and shoes and boots [20]. This might be related to the lifetime of clothes with an average life span of

5.4 years [22] and varies over categories; for example, T-shirts and jeans have a life span of 3–4 years [8].

This research shows that 25% of the items in the wardrobe are unused. This is lower than the data from the study by Maldini et al., where the average percentage of unused garments was 28% in the Netherlands and 30% in Germany [20]. This might be a result of the larger sample size in this study, the international sample or again the method of data collection. In any case, the 25% of unused items in the wardrobe systems present a vast amount of untapped resources in terms of garments that might be suitable for reuse, repair or recycling that are now trapped inside our wardrobes.

On average, 17% of the total wardrobe consists of second-hand items. This average is much higher than the compared study by Maldini et al., which measured 6% in the Netherlands in 2017 [20]. Possibly the second-hand clothing market has grown in the last few years, where second-hand garments are more common nowadays. On the other hand, the participants in the research by Maldini et al. used snowball sampling, which might have resulted in a non-representative sample less interested in second-hand clothing [20]. The respondents of the survey in this study have decided to follow the Circular Fashion MOOC, and therefore based on their interests and values might already be disposed to sustainable and circular behavior which could have translated to their purchasing actions, choosing second-hand garments more often. More research with a much larger representative sample of the different countries from a broader audience would be needed to understand whether the percentage of second-hand items in wardrobe systems indeed has increased due to more sustainable behavior, or if the high percentage of this study is a result of the predisposed interests of the respondents.

Regarding the age differences, it is has been found that respondents in the category 51+ have more clothing items, compared to the other categories (18–30 and 31–50), which is incongruent with Maldini et al., who mentioned that on average younger people (18–30) owned more apparel compared to older participants (30+) [20]. Fast fashion is a phenomenon that is often prescribed to the buying and dispensing behavior of younger generations. As this study does not allow for research within the clothing categories to brand origin, it is not possible to connect these results to fast fashion purchasing behavior of respondents in different age categories. More research would be needed in order to assign fast fashion as a cause for accumulating garments and wardrobe inflow. Naturally, to make this case, wardrobe outflow should also be investigated, which is currently not part of the survey in this study. However, it remains remarkable that the data from this study show different results, as it was not the youngest category (18–30), but the oldest category (51+) who owned the most clothing items. Whether that is a result of, for example, the total amount of time to accumulate clothing items, connecting more value to their clothing items and/or purchasing power of this category is unsure.

In addition to owning the most clothing items, respondents in the category 51+ also had the most unused and second-hand items in their wardrobe. This is also incongruent with the data from Maldini et al. [20]. More insight is needed in the wardrobe of this age group to understand their behavior.

The number of unused clothes and second-hand is related to wardrobe size, as is seen in the results of both this study and in the study by Maldini et al. [20]; however, these results were found in different age categories, respectively. Having a smaller wardrobe size reduces the number of unused garments, and therefore lowers the unused fashion and textile resource potential in wardrobes and the fashion industry. However, it also reduces the number of second-hand items, which in a circular fashion industry are highly needed.

Differences between nationalities are only found when looking at the total amount of second-hand clothes. French, Spanish and Indian respondents indicated to own the least second-hand clothing items compared to respondents from the Netherlands, the United Kingdom and the United States. However, this provides an opportunity for more research to understand the barriers of second-hand fashion consumption and targeted interventions

for circular fashion behavior in France, Spain and India. Possibly these countries can learn from motivations in the other countries where amounts of second-hand are higher.

There were no significant differences found between the countries for the total amount of clothing items, nor the amount of unused clothing items. Therefore, interventions for circular behavior based on reducing the total amount of clothing items that consumers own, as well as interventions based on reducing the amount of unused clothing items, could be deployed in all countries researched in this study.

### 4.2. Measuring and Monitoring Clothing Systems

This study is an attempt to acquire data on wardrobe systems internationally and to gain insight into current behavior to target interventions for a circular fashion industry. In that sense it is also the first attempt to develop a digital monitoring survey to make data collection of a wardrobe easier in terms of time and effort, allowing for more data collection through a larger sample. However, this type of data collection also presents some challenges. As respondents fill in the survey without supervision, the survey itself needs to reduce any room for (mis)interpretation and error. This research has shown that some changes would be required of the format of the digital survey to gather valuable data and include as many responses as possible.

1. Add more clothing categories to the survey and adapt it to international respondents, first by adding different cultural clothing categories as predefined options instead of the blank category as in this study that respondents could fill themselves. Additionally, include socks, underwear, sportswear and swimwear.
2. Answer possibilities in digital survey, use a drop-down menu instead of allowing respondents to type their answer, to avoid answers such as, for example, only mentioning "yes" or "no" or saying "20+" items.
3. Forced response in the digital survey, to understand whether respondents simply do not own items of a certain category or did not/forget to fill out the category.
4. Measure the time taken to fill in the survey.
5. Add questions related to buying and disposal to create an overview of the entire wardrobe system, including inflow and outflow.
6. Add questions, to create insight in the "motivations? why" of the behavior and understand the values behind the garments within the wardrobe system.
7. Add other demographic information, urban–rural, income, education, study background.

In addition to changes to the survey, it is also suggested (and in line with Klepp and Bjerck) that it would be of great benefit to the study to combine the method with other data collection and information-gathering methods to get a better understanding of the reasoning behind the amounts of (un)used and second-hand clothing [12].

### 4.3. Recommendations for Reuse Interventions

Insight into the number of garments, (un)used and second-hand is necessary to formulate interventions to increase reuse and move garments from practical service life to technical service life. For all interventions targeting is needed for example based on motivations and level of awareness, knowledge and understanding linked to the possibilities consumers have in their daily life situation. In Table 7 we propose more detailed interventions to support reuse with regard to ownership and washing practices. These can also be aimed at specific target groups, those who already are and are not familiar with second-hand or renting/leasing clothes, age groups as well as related to different clothing categories.

**Table 7.** Interventions for circular reuse/resell behavior for consumers.

| R-Strategy | Consumer Behavior | Sources of Behavior | Intervention |
|---|---|---|---|
| Reuse/ resell | Reusing (second-hand) items | Motivation | Raise awareness of amount of (unused) garments before new items are bought, especially for those categories which are often unused, e.g., short-sleeve T-shirts and tops, shoes and boots, dresses and shawls and scarves. |
| | | Capability | Inform how to recognize sustainable garments. Provide tools and assistance to help consumers understand their preferred style and cuts that suit their body shapes [10]. Educate consumers to recognize quality in garments [23,24] Include textile skills in school curriculums [10]. |
| | | Opportunity | Provide possibilities of where to buy second-hand or more sustainable clothing. Support social acceptance of second-hand clothes. Create opportunity for specific target groups to experiment with reuse and particular second-hand garment categories based on their level of experience and acceptance. Provide repair services [10]. |
| | Better washing practices | Motivation | Improve awareness of time, money and labor savings from reduced frequency and temperature of washing garments [10]. |
| | | Capability | Provide tools and assistance to help consumers to reduce washing frequency and other sustainable washing behaviors. |
| | | Opportunity | Legislate availability of sustainable detergents/restrict harmful detergents. Provide washing guide relating to garment label care instructions. |
| | Leasing/renting garments | Motivation | Raise awareness and create acceptance of leasing and renting garments. |
| | | Capability | Inform what kind of clothes could ideally be rented or leased. |
| | | Opportunity | Provide possibilities where to lease and rent garments [10]. |
| | Selling/donating garments for reuse | Motivation | Raise awareness of the need to those consumers with a lot of unused garments to the benefits of selling or donating garments. |
| | | Capability | Inform consumers where they could sell or donate their unused garments. Upcycling [10]. |
| | | Opportunity | Provide the possibilities where garments can be collected. Legislate clothing recycling [10]. Provide recycle services [10]. |

Note: Interventions targeting the motivation of consumers have been specified per circular consumer behavior. However, they all can be addressed through one of the interventions proposed by Harris et al. [10] "social marketing campaigns" to improve awareness and therefore impact consumers' motivation for change. This intervention is therefore not separately mentioned in the table.

### 4.3.1. Reusing (Second-Hand) Items

Awareness campaigns for reducing buying of garments could focus on categories that are currently most unused, short-sleeve T-shirts and tops, shoes and boots, and dresses. Additionally, relative numbers show that shawls and scarves are the most unused and should therefore be included.

For improving capability, it is recommended to inform consumers about sustainable fashion. Additionally, according to Harris et al. it is needed to provide tools and assistance to help consumers understand their preferred style and cuts that suit their body shapes [10]. This should improve decision-making whilst purchasing but also for actions based on reuse/resell behavior such as deciding to keep, donate or resell garments. A better understanding of the quality of garments [23], coupled with a willingness to invest in higher-quality garments, as opposed to fast fashion, has the potential to extend the lifespan of garments [24] and therefore improve reuse. Lastly, including textile skills in school curriculums is another recommendation by Harris et al. [10], which will teach mending or

adjustment practices in order to elongate the use of garments and elongate their practical service life.

Interventions targeting the opportunity for circular consumer behavior reuse/resale focus on second-hand garments, and elongating use of garments through repair. Currently, second-hand items in wardrobes are mostly (1) short-sleeve T-shirts and tops, (2) coats and jackets and (3) sweaters and cardigans. For more experienced wearers of second-hand items (females), improving acceptance of reusing other clothing categories is recommended. For reusing (second-hand) items overall, males have lower numbers of second-hand clothing, and improving the motivation, capabilities and opportunity of reuse and second-hand garments for this group would be recommended. The same goes for wearers with an age between 31–50. Thus, for wearers with less experience with second-hand items (males and/or people aged 31–50) an introduction to reuse in these categories ((1) short-sleeve T-shirts and tops, (2) coats and jackets and (3) sweaters and cardigans) might stimulate more behavior as it is more widely accepted already. In addition, it is recommended to provide ample opportunity to repair garments, which as, Harris et al. propose, can be achieved through retailers [10].

### 4.3.2. Better Washing Practices

"It is well known in the textile literature that the laundering process degrades clothing, but this knowledge may not extend to consumers who are less conscious and aware of how cleaning can impact the longevity of their clothing" [25] (p. 42). The interventions and recommendations in Table 7 for better washing practices are based on Harris et al. [10] as this was out of scope in this research, but washing less frequently and therefore extending the practical service life of garments is a proven way to improve sustainability in the fashion industry [25].

In terms of improving motivation, from the research by McQueen et al. specifically focusing on use and washing practices of denim can be concluded that motivations for reducing washing frequency should combine environmental factors (reduced energy and water) as well as knowledge that the garment will last longer [25], a recommendation that is in line with the study by Laitala and Klepp [26].

### 4.3.3. Leasing/Renting Garments

This behavior intends to move away from consumption and reliance on resources towards service-oriented consumption. "The potential to utilize services to support long-term use of and engagement with clothing products to enhance sustainability holds much promise" [27] (p. 19). As Armstrong et al. put it, "consumers become married to existing solutions and socio-cultural regimes, making the implementation of more radical concepts more difficult to accept" [27] (p. 21). As leasing/renting garments is not yet fully adopted by consumers, interventions to improve awareness and social acceptance are essential for these behaviors to become more widely implemented. To improve leasing/renting, it is recommended that the focus should be on portraying an attractive lifestyle and experiencing personal style, creativity and change through leasing/renting [27].

Interventions for improving consumers' capability of leasing/renting garments can focus on increasing the understanding and opportunity of leasing/renting items that are currently mainly unused in the wardrobe, such as short-sleeve T-shirts and tops, dresses and shawls and scarves. Shoes and boots have been identified as an important category, and despite being the second-largest unused category they seem less adequate for a leasing/renting model.

Females (especially the oldest age group, 51+) have more unused items, so the leasing/renting models can have a large impact for this group and targeting the leasing/renting interventions on motivation, capability and opportunity to this group first would be recommended.

### 4.3.4. Selling/Donating Garments for Reuse

Almost 25% of the garments in the wardrobe are unused. When determining which items are at the end of their life, a distinction is made between absolute obsolescence (product failure) and relative obsolescence [24], which could mean "the replacement of a usable product due to improved function of a newer model, changes in fashion or personal style or when products have been intentionally designed to wear out after a short period of time" [24] (p. 587). As Degenstein et al. describe, the type of obsolescence, which is determined by the consumer, is important for the disposal method and lifespan of the garments [24]. Their study shows that the price of purchase is important and that more expensive designer garments are more likely to be mended, take more effort to dispose of and are more likely to be reused [28] than cheaper (fast-fashion) items [24]. Research by McNeill et al. finds that used garment life extension is influenced overall by garment damage and perceived quality, as well as by garment type in some scenarios [28]. In addition, the personal attachment of people and their garments is another factor in deciding to postpone disposal of their garments. This is described by Niinimaki and Armstrong as embodied memories in person–product attachment [29]. Therefore, the motivation of consumers to sell/donate/dispose their items is multilayered, so the interventions for motivations should be as well. Unfortunately, this study only covers the items in the wardrobe that are unused and not disposed of (yet), without further insight into the motivation explaining that behavior.

Overall, it is recommended that circular interventions for selling/donating garments for reuse should improve motivation, capability and opportunity first and foremost for these unused items to be reused (through repair for example), otherwise to be donated for reuse or lastly to be refurbished/remanufactured through upcycling. If wearers are able and willing, they can do this themselves, or opportunities should be improved for others to make use of these otherwise unused resources to allow reuse of garments (as a resource). Such opportunities could be provided by retailers, as mentioned by Harris et al. [10], but other forms of donating/recycling should be stimulated as well. All in all, opportunities for donating garments for reuse and recycling naturally depend on national legislation for clothing recycling and should be made available as mentioned by Harris et al. [10].

### 4.4. Limitations

The non-representative sample of the Circular Fashion MOOC might have attracted learners with a higher interest in clothing and sustainability than average, which might on one hand lead to having more clothes overall due to an interest in textiles and fashion, or could result in more conscious fashion consumption and behavior, resulting in a lower number of clothes in total or possibly more second-hand clothing items.

The respondents in this study have filled in the survey individually, also called a self-report study, which is an inexpensive and more simple data collection method allowing data collection in a relatively short time and reducing the influence from an interviewer's interpretation or behavior. However, it also has pitfalls for example as respondents might want to give socially acceptable answers or none of the answer possibilities seem to fit their answers, as could be seen in the predefined clothing categories in the survey.

The digital wardrobe study, as developed as an assignment in the Circular Fashion MOOC and used as data collection for this study, has allowed for both quantitative and qualitative input from respondents. The quantitative measurements were gathered through the digital survey and were suitable for data analysis. The qualitative input consisted of posts in a discussion forum within the course and were unfit for qualitative analysis. Therefore, this study was limited to quantitative data analysis, and improvements on the qualitative data gathering within a digital wardrobe study are needed.

The digital wardrobe study approach as developed could be a valuable tool for monitoring the amount of clothes and behavioral changes if it could be measured longitudinally. The Circular Fashion course is usually undertaken by new learners each course run, and therefore does not allow for longitudinal study of the same respondents. There would be

value in both measuring and monitoring changes within respondents' wardrobes as well as new samples to discover trends and behavior changes over time.

Furthermore, this way of measuring and monitoring wardrobes is not carried out that much, meaning that this study can be used as an experiment in developing and improving the right tool or method to measure what is inside people's wardrobes on a large scale.

### *4.5. Future Research*

#### 4.5.1. Future Research Exploring Wardrobes

Further research would be necessary to better understand the behavior and motivations of inflow and outflow of the wardrobe in relation to age and gender to understand the values of clothing items, and designing circular interventions to stimulate more circular fashion behavior.

Understanding age differences and clothing accumulation, with a focus on the oldest age category understanding their behavior based on time; to accumulate clothing items, emotion; connecting more value to their clothing items, and/or purchasing power requires more research.

Understanding not only the number of garments in the wardrobe but gaining more overall insight in the types of garments within the wardrobe, such as quality, fast-fashion brands, thrifted items, family heirlooms, etc., is important. This research would be needed among others to investigate in what gender, age groups or clothing categories in the wardrobe fast fashion plays a role, and whether fast fashion is a cause for accumulating garments, and to understand the role it plays more generally regarding wardrobe inflow and/or outflow.

Data in this study have shown only differences between nationalities based on second-hand clothing items. More research with a more representable sample is needed determine the value of these results. Additionally, research is needed to understand the barriers of second-hand fashion consumption between the countries, as well as targeted interventions for circular fashion behavior to prioritize improving second-hand fashion consumption in the countries that have scored the lowest, in this study France, Spain and India.

#### 4.5.2. Future Research for Measuring and Monitoring

The teachings of this research focus mainly on the quantitative results of the data collection, and therefore provide insight into the number of garments that are (un)used and second-hand, which give insight into potential interventions for behavior. However, there is a lack of understanding of the motives for the behavior and insight in the reasoning behind the number of garments. Therefore, to gain better insight into the number of garments and the motivation behind inflow or outflow of the wardrobe, it is needed to complement the quantitative data analysis with alternative methods for data gathering. Standard methods such as interviewing or observing, as proposed by Harris et al. [10] and executed by Maldini et al. [20], are less suitable for collecting data from a large international sample online. Therefore, other methods should be investigated. A simple improvement would be the addition of qualitative open-ended questions in the survey, but also alternative methods should be explored, for example the use of photos of the wardrobe or specific items.

In addition, this research has been conducted based on a non-representative sample of mainly women (70%) and respondents mostly from Europe (47.9%). Therefore, it is recommended to gather data from a representative sample and appropriate sample size, as well as equal distribution of female and male and nationalities in future research.

#### 4.5.3. Future Research for Interventions

The detailed overview of interventions, structured according to buying, using and disposal in terms of consumer behaviors and sources of behavior, will be worthwhile to test whether there might be impact on reuse for different target groups, clothing categories and different countries.

## 5. Conclusions

Amounts of (un)used and second-hand garments are explored and provide insights to target interventions in which attention is paid to several behaviors of consumers, including how to support their motivations, capabilities and opportunities. Thus, by means of a better understanding of wardrobes and its monitoring, recommendations are provided to support the reuse of consumers' clothing, aiming towards a more circular clothing system.

**Author Contributions:** Conceptualization, D.d.W.; J.G. and S.J.S.; methodology, J.G. and D.d.W.; formal analysis, J.G. and D.d.W.; writing—original draft preparation, D.d.W. and S.J.S.; writing—review and editing, D.d.W. and S.J.S. All authors have read and agreed to the published version of the manuscript.

**Funding:** This paper is part of the project Recycling and end-of-life strategies for sustainability and climate (KB-34-011-001), financed by the Dutch Ministry of Agriculture, Nature and Food Quality.

**Institutional Review Board Statement:** Ethical review and approval were waived for this study, due to the privacy policy of edX. Users of the platform agree that edX and members use information, including personal information, to support scientific research.

**Informed Consent Statement:** Patient consent was waived due to the reason that the data set with which this research has been conducted, cannot be traced back to traceable personal data. This research used anonymous data.

**Data Availability Statement:** The data presented in this study are available on request from the corresponding author. The data are not publicly available due to privacy.

**Conflicts of Interest:** The authors declare no conflict of interest. The funders had no role in the design of the study; in the collection, analyses or interpretation of data; in the writing of the manuscript or in the decision to publish the results.

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
