# Peer review of "Exploring Worldwide Wardrobes to Support Reuse in Consumers’ Clothing Systems"

_sustainability, doi:10.3390/su14010487_

Round 1

Reviewer 1 Report

The authors have embarked on a very interesting topic. I enjoyed reading the manuscript thoroughly. However, some minor amendments are suggested to improve the quality of the manuscript.

For example, gap in the literature is not properly mentioned. It would be good if the authors could explain how this research differs from other studies in the field.   

The authors used convenience sampling whereas in the literature there are drawback mentioned on this particular sampling technique. As such, it is suggested to justify the choice sampling technique.

Justification for the choice of the sample size is missing.

Almost 70% respondents are female which requires justification.

Author Response

Thanks for your review. We have taken a careful look, please see attachment. 

Reviewer 2 Report

Exploring Worldwide Wardrobes, to support reuse in consumers’ clothing systems

ID:Sustainability-1510491

Review comments

            The manuscript discusses the consumer wardrobe study with some international consumers mainly Europe based consumers who attended some online course.  First of all I would like to congratulate the authors for working in a needed area. Further, it is important to conduct such study among international consumers is one of the need of the hour. However, I would like to suggest following points to improve the manuscript.

  1. To get some clear picture on the questions and options provided, I could not found any questioner (or appendix) at the end of paper. Inclusion of the questioner is essential for the readers understanding.
  2. Without those questions and the option provided in. It is difficult to interpret the results from my side.
  3. Section 4.2 mentioned the corrections required in the survey questions – do these correction incorporated in the current survey? If so why it is provided on the manuscript.
  4. If not why this section (4.2) not considered before planning the survey?
  5. In line 315 – 316 : Author reported that elderly people used more cloths then the younger people. This statement need to be validated with few literature. This information might be influenced by the higher participation from Europe.
  6. Line 373 – 374 – What is that other data collection method you wish to convey here??
  7. Is this method used in this study? If not, why it is not considered?
  8. In the first part of the paper, is very well written. But in the second part, especially in the intervention section, most of the information reported were from Harris et al. [11]. Here author’s own interpretation is less. For example Table 7.
  9. Line 404 – 408: in which author discussed about the quality of the garment – However, I personally feel, even a quality garments are disposed due to the fast fashion trend (which is mentioned in Line 309). But author did not mention about the fast fashion and its impact on fast disposal of clothing. Inclusion of those factors also adds values to the interventions discussed.
  10. Line 429 – Reference Harris et al. should be 11, it is marked as 10.
  11. IN section 4.3.1 – while talking about the second hand garments, it is important to mention how to obtain that second hand clothing – is it from thrift stores or it is form family members?
  12. 3.2 – washing practices seems bit irrelevant to the interventions that are detailed in this manuscript.

Incorporation of abovementioned correction will improve the quality of the manuscript.

Overall, the manuscript gave an impact of different clothing (used and unused) in the participants wardrobe. Based on that, authors discussed lot of intervention methods to improvise. However, I feel, there is a lack of details provided on the implementation part. Secondly, Out of total participation, 47.9% participants from Europe, this might influenced the results as the other continental participations seems very less compared to this. Providing solution to this issue (either by literature support or by authors justifications) and solving the mentioned issues will improve the quality of the manuscript.

Author Response

(The authors gave the same response as above.)
